# Cervical cancer screening in Brazilian Amazon Indigenous women: Towards the intensification of public policies for prevention

**Iria Ribeiro Novais**, **Camila Olegario Coelho**, **Helymar Costa Machado**, **Fernanda Surita**, **Luiz Carlos Zeferino**, **Diama Bhadra Vale** *

Department of Obstetrics and Gynecology, University of Campinas, Campinas, São Paulo, Brazil

* dvale@unicamp.br

## Abstract

### Background

Indigenous women are vulnerable to cervical cancer. Screening is a strategy to reduce the burden of the disease.

### Objective

To evaluate the prevalence profile of cervical cancer screening cytological results in Brazilian indigenous women by age and frequency of tests compared to non-indigenous women.

### Methods

A cross-sectional study evaluating the prevalences of screening test results in indigenous women assisted in the Brazilian Amazon from 2007 to 2019 (3,231 tests), compared to non-indigenous women (698,415 tests). The main outcome was the cytological result. Other variables were frequency, age groups, and population. The frequency was categorized as "1st test", the first test performed by the women in their lifetime, or "screening test," tests from women who had previously participated in screening. Analyzes were based on prevalences by age group and population. We used Prevalence Ratios (PR) and 95% Confidence Intervals for risks and linear regression for trends.

### Results

Data from the 1st test showed a higher prevalence of Low-grade Squamous Intraepithelial Lesion (LSIL) in indigenous women. Peaks were observed in indigenous under 25, 35 to 39, 45 to 49, and 60 to 64. The prevalence of High-grade Squamous Intraepithelial Lesion or more severe (HSIL+) was low in both groups in women younger than 25. The indigenous HSIL+ prevalence curve showed a rapid increase, reaching peaks in women from 25 to 34 years, following a slight decrease and a plateau. In screening tests, HSIL+ was more prevalent in indigenous from 25 to 39 (PR 4.0,2.3;6.8) and 40 to 64 (PR 3.8,1.6;9.0). In indigenous, the PR of HSIL+ results in screening tests over 1st tests showed no screening effect in

**Data Availability Statement:** Metadata is available at https://doi.org/10.25824/redu/9BLFCK repository. Original indigenous data contain

sensitive information. For availability under reasonable request, it needs Ethics Committee authorization (CONEP – Brazilian agency which regulates research on vulnerable populations, conep@saude.gov.br).

**Funding:** The author(s) received no specific funding for this work.

**Competing interests:** The authors have declared that no competing interests exist.

**Abbreviations:** ASC, Atypical Squamous Cells; ASC-H, which cannot exclude high-grade intraepithelial lesion; ASC-US, ASC of undetermined significance; CI, 95% Confidence interval; EDS, The non-profit Organization "Expedicionários da Saúde"; HPV, Human Papillomavirus; HSIL+, High-grade Squamous Intraepithelial Lesion or more severe—Squamous Cell Carcinoma, Atypical Glandular Cells, Adenocarcinoma in situ and Invasive Adenocarcinoma; LEEP, Loop electrosurgical excision procedure; LSIL, Low-grade Squamous Intraepithelial Lesion; PHC, Primary health care; PR, prevalence ratio.

all age groups. In non-indigenous, there was a significant effect toward protection in the age groups over 25.

## Conclusion

This screening study of indigenous women from diverse ethnicities showed a higher prevalence of cytological LSIL and HSII+ than in non-indigenous women. The protective screening effect in reducing HSIL+ prevalence was not observed in indigenous.

## Background

Indigenous women in Latin America are more vulnerable to cervical cancer than non-indigenous women. The prevalence of precursor lesions is high, cases are diagnosed in advanced stages, incidence and mortality rates are high, and survival is low [1–14]. Despite the variable rates of HPV prevalence, genetic diversity is reported and overlaps with poor knowledge regarding the immunological aspects [2, 8, 11, 13, 15].

The origin of those women points to a diverse ethnicity related to their geographical location and family group origin, which in turn reflects in their sexual reproductive behavior. A central issue common to all groups is the fragility of healthcare access, leading to a high prevalence of diseases influenced by social determinants, such as infectious disease, infant mortality, violence, and cervical cancer [16].

Brazil is estimated to be around one million indigenous people, one-third of whom live in the Amazon region. There are more than 300 ethnicities. The non-profit Organization "Expedicionários da Saúde" (EDS) provides medical care to geographically isolated indigenous populations in the Brazilian Amazon [17]. Since 2004 expeditions have been made in partnership with the Indigenous Secretary of the Ministry of Health. One of the healthcare actions performed is cervical cancer screening through cytology smears, following the Brazilian official recommendation [18]. The slides are processed and analyzed in the cytopathology laboratory of the Women's Hospital of the University of Campinas (CAISM/Unicamp).

This study aimed to analyze the prevalence of abnormal results in cytologic screening tests of indigenous women from the Brazilian Amazon from 2007 to 2019, compared with non-indigenous women. The expectations were to provide a comprehensive profile of high-risk results prevalence by age and to evaluate the screening efficiency. The diversity of ethnicities analyzed may present a broad understanding to support the implementation of the screening recommendations.

## Methods

It was a cross-sectional study evaluating the prevalence of cytological results from cervical cancer screening in indigenous women assisted by a non-profit organization in the Brazilian Amazon from 2007 to 2019. Cytological results were recorded in the Women's Hospital laboratory database of the University of Campinas, the data source. We compared the indigenous results with tests recorded in the same database from 2007, 2011, 2015, and 2019. The subjects were called "indigenous"–women attended by the organization in remote areas of the Amazon rainforest (S1 Table), and "non-indigenous"–women from the general population using the health care facilities of Campinas and surroundings, whose cytological results were stored in the same lab. The total sample was 3,513 tests from the indigenous and 769,729 tests from non-indigenous women.

Cervical cancer prevention in Brazil is based on vaccine and screening actions, both free of charge in the public health system. The vaccine program targets girls and boys from 9 to 14 years old in the two doses schedule (0/6) [19]. Although the coverage in the overall population is around 50% [19], indigenous people's coverage was over 85% before the pandemic (data available from the Ministry of Health upon request). Overall, screening is cytology-based through an opportunistic program (no personal invitation) targeting women 25 to 64 years old every three years at primary health care (PHC). Precursor lesions are treated in referral facilities by LEEP [18]. Coverage is low in indigenous and non-indigenous people, usually higher following the socio-economic level of the region. Indigenous women may be accessed to cytology by local PHC or non-profit organizations. There is no specific recommendation for indigenous women regarding screening [18]. Indigenous women are unlikely to be treated in their place of origin, especially when living in remote areas of the Amazon rainforest, requiring a complex and articulated framework to access a referral facility.

The non-profit organization EDS has carried out two or three annual expeditions to the Brazilian Amazon since 2004, performing low-complexity surgical procedures and attendance in gynecology, obstetrics, pediatric, general surgery, orthopedy, and ophthalmology. Women's health care includes cervical cancer screening, and some expeditions treat precursor lesions by LEEP. Those actions complement the fragile local opportunistic screening by PHC in the region. Test results were returned to PHC, and abnormal results followed the regular multi-level pathway throughout referral regional facilities to colposcopy and treatment, including those results suggestive of invasive lesions. In indigenous women, only one test suggested invasive lesion.

The University of Campinas laboratory analyzes cytology primarily from women who had their test collected in the public health system of Campinas region, comprising a population of nearly three million inhabitants. It is certificated as a high-quality cytology laboratory and performs quality control for other laboratories in São Paulo state. It stores information about the women and the cytological results. The professional who collects the test (physician or nurse) records the clinical data using a form that is submitted to optical reading. The cytology results prevalences are similar to those identified nationally by the central Ministry of Health database [18].

All EDS tests identified in the database from 2007 to 2019 were included. Totals were confronted with documents from the expeditions to corroborate their origin. For comparison, all other tests from the database in 2007, 2011, 2015, and 2019 were included. The exclusion criteria were tests from women previously submitted to hysterectomy; tests collected for purposes other than screening (follow-up of previous abnormal result or post-treatment procedure); unsatisfactory smears according to the Bethesda criteria [18]; and those without information on age or result. In the original collection form, "frequency" was recorded as "first test," "less than one year," "one year," "two years,". . ., "five years," and "more than five years." After exclusion criteria were applied, in the category "frequency," those with unknown "frequency" were redistributed by age group and result, and then excluded those reported as "less than one year" because they were considered as "tests collected for purposes other than screening."

The outcome variables were the cytological results. Cytopathological reports used in the laboratory follow the Brazilian Nomenclature for Cytopathological Reports [18], based on the Bethesda System. In this study, we categorized results as "Low-grade Squamous Intraepithelial Lesion (LSIL)" or "High-grade Squamous Intraepithelial Lesion or more severe" (HSIL+). HSIL+ corresponded to Squamous Cell Carcinoma, Atypical Glandular Cells, Adenocarcinoma *in situ*, and Invasive Adenocarcinoma. Because of their low frequency, we grouped those results into one single category. The totals included the results "negative for neoplasia" and "Atypical Squamous Cells (ASC-US–of undetermined significance or ASC-H–which

cannot exclude high-grade intraepithelial lesion)". ASC-H was not included as HSIL+ because the database merges all ASC in one category (the categorization as ASC-US or ASC-H is presented written in the original form).

The other variables were frequency, age groups, and population (indigenous and non-indigenous). The frequency was categorized as "1st test", the first test performed by the women, or "screening test", tests from women who had previously participated in screening (non-1st test). It is important to note that the data from the 1st test indicates the natural history of the prevalence in that population since they represent tests in women who were not submitted to previous screening/treatment practices. Thus, assuming a high specificity of cytological results, the "1st test" is a proxy of the prevalent lesions in that population. Screening tests indicate the prevalence in women who may have or may not have been submitted to procedures in the cervix that can disrupt the natural history of the disease. In the screening context, a protective effect is expected to reduce the prevalence of precursor or invasive lesions (HSIL+) [20–22].

Statistical analyzes were based on the prevalence of results by age group and population (indigenous and non-indigenous). Prevalences were calculated using as a numerator the number of tests by each result by age group (LSIL or HSIL+), and as a denominator the total number of tests in that age group (any result). The results were presented per 10,000 tests. Prevalence Ratio (PR) and their respective 95% Confidence Intervals (CI) were calculated to relate the variables of interest. Trends were assessed by linear regression. The power of the study was calculated by a significant level of 5% in the group of women in their 1st test. It was used "The SAS System for Windows (Statistical Analysis System), version 9.4." (SAS Institute Inc, 2002–2012, Cary, NC, USA).

The project was approved by the "Research and Ethics Committee of the University of Campinas" and by the "National Committee of Ethics in Research" (CONEP) under the number CAAE: 51446421.9.0000.5404. Both Committees waived the need for Informed Consent due to the study's retrospective nature.

## Results

From the total number of tests, were excluded respectively in indigenous and non-indigenous women: "hysterectomy" 19 and 21,231 tests; "tests collected for purposes other than screening" 222 and 60,780 tests; "unsatisfactory" 38 and 7,394 tests; unknown "age" 7 and 418 tests; and unknown "result" 38 and 7,878 tests. More than one exclusion criterion could be applied to each test.

The final sample consisted of 3,231 cytology from indigenous women and 698,415 from non-indigenous women: respectively, 1,162 and 55,954 from women performing the test for the first time (1st test), and 2,069 and 642,461 from women who have already participated in screening (screening test). The power of the study to detect a difference in HSIL+ in their 1st test was 97.6% (0.89% in indigenous versus 0.26% in non-indigenous, β = 0.976). The description of the expeditions and ethnicities covered are displayed on the S1 Table.

Table 1 shows the proportions of the aggregated results (LSIL, HSIL+, and totals). Overall, respectively in indigenous and non-indigenous women, the proportion of tests performed in women younger than 25 was 21.7% and 19.4%; from 25 to 39 years 43.1% and 35.7%; from 40 to 64 years 31.6% and 40.4%; and over 64 years 3.6% and 4.6%. The proportion of LSIL results was 0.9% in indigenous and 0.4% in non-indigenous. Of HSIL+ results, it was 0.9% in indigenous and 0.3% in non-indigenous.

We analyzed the PR of LSIL and HSIL+ cytological results in indigenous and non-indigenous women in their 1st test (Table 2) and their screening test (Table 3). Data from the 1st test showed a higher prevalence of LSIL in indigenous women than non-indigenous women: in

**Table 1. Proportion of cervical cancer screening cytological results by age groups in indigenous (3,231 tests) and non-indigenous women (698,415 tests).**

| Age group | LSIL | | HSIL+ | | Total (Neg+ASC+LSIL+HSIL+) | |
|---|---|---|---|---|---|---|
| | Indig | Non-indig | Indig | Non-indig | Indig | Non-indig |
| < 25 | 2.1% | 1.0% | 0.1% | 0.3% | 21.7% | 19.4% |
| 25 to 39 | 0.8% | 0.4% | 1.4% | 0.3% | 43.1% | 35.7% |
| 40 to 64 | 0.5% | 0.1% | 0.7% | 0.2% | 31.6% | 40.4% |
| > 64 | 0.0% | 0.0% | 0.8% | 0.3% | 3.6% | 4.6% |
| Total | 0.9% | 0.4% | 0.9% | 0.3% | 100.0% | 100.0% |

Legend—< 25: women under 25; > 64: women over 64; Neg: Negative; ASC–Atypical squamous cell; LSIL: Low grande intra-epithelial lesion; HSIL+: High-grade intra-epithelial lesion or more severe; Indig: Indigenous women; Non-indig: Non-indigenous women.

younger than 25, the PR was 2.0 (CI 1.1;3.6); in 40 to 64 was 9.6 (CI 2.7;34.1); and in older than 64 lower than zero. The prevalence of HSIL+ was low in both groups in women younger than 25. In women aged 25 to 39, the PR of indigenous versus non-indigenous was 3.0 (CI 1.3;6.8). In the oldest age groups, the prevalence in indigenous women was lower, although not significantly (Table 2).

Regarding the screening tests, LSIL was more prevalent in indigenous women under 25, although not significantly. HSIL+ prevalences were zero in indigenous younger than 25 and older than 64. The prevalences in indigenous were higher than in non-indigenous women from 25 to 39 (4.0, CI 2.3;6.8) and 40 to 64 (3.8, CI 1.6;9.0) (Table 3).

Table 4 presents data on the risk of women who had already participated in screening (screening test) or who were performing their 1st screening test (1st test). No screening effect could be observed in indigenous except for LSIL in women aged 40 to 64 (PR 0.05; CI 0.00;0.81). In non-indigenous, a screening effect toward protection was observed for HSIL+ in women older than 25, more pronounced in women older than 40: PR 0.19 (CI 0.13;0.27) in women aged 40 to 64; and 0.10 (CI 0.06;0.15) in women older than 64.

Figs 1 and 2 show the trend curves for the prevalences by age groups regarding their 1st screening test (1st test). In Fig 1 it is possible to compare the LSIL and HSIL+ curves in indigenous versus non-indigenous women, and in Fig 2, the LSIL and HSIL+ curves in each population group. In both LSIL curves, a trend to decrease the prevalence with aging was observed, although significant only in non-indigenous women (p<0.001). The trend was not significant in indigenous women, but peaks of LSIL prevalence were observed in women under 25 and in the age groups of 35 to 39, 45 to 49, and 60 to 64. In both HSIL+ curves, there was a trend

**Table 2. Prevalence ratio of cytological results comparing indigenous and non-indigenous women in their first cervical cancer screening test (1,162 and 55,954 tests, respectively).**

| | LSIL | | | HSIL+ | | |
|---|---|---|---|---|---|---|
| | Indig | Non-indig | | Indig | Non-indig | |
| Age group | P (/10,000) | | PR (95% CI) | P (/10,000) | | PR (95% CI) |
| < 25 | 264.8 | 131.3 | 2.0 (1.1;3.6) | 24.0 | 17.3 | 1.4 (0.2; 9.9) |
| 25 to 39 | 146.5 | 72.3 | 2.0 (0.9;4.8) | 169.8 | 56.5 | 3.0 (1.3;6.8) |
| 40 to 64 | 149.5 | 15.4 | 9.6 (2.7;34.1) | 84.4 | 97.3 | 0.9 (0.2;3.2) |
| > 64 | 0.0 | 0.0 | - | 134.9 | 233.4 | 0.6 (0.1;4.2) |

Prevalence ratio–indigenous/non-indigenous women.

Legend—< 25: women under 25; > 64: women over 64; LSIL: Low grande intra-epithelial lesion; HSIL+: High-grade intra-epithelial lesion or more severe; Indig: Indigenous women; Non-indig: Non-indigenous women; P–Prevalence; PR: Prevalence ratio; CI–Confidence Interval).

**Table 3. Prevalence ratio of cytological results comparing indigenous and non-indigenous women attending cervical cancer screening (2,069 and 642,461 tests, respectively).**

| | LSIL | | | HSIL+ | | |
|---|---|---|---|---|---|---|
| | Indig | Non-indig | | Indig | Non-indig | |
| Age group | P (/10.000) | | PR (CI 95%) | P (/10.000) | | PR (CI 95%) |
| < 25 | 134.8 | 86.5 | 1.6 (0.6;4.2) | 0.0 | 28.7 | 0.6 (0.0;9.8) |
| 25 to 39 | 53.0 | 41.0 | 1.3 (0.6;3.0) | 131.5 | 33.1 | 4.0 (2.3;6.8) |
| 40 to 64 | 7.9 | 14.2 | 0.6 (0.0;7.3) | 70.7 | 18.5 | 3.8 (1.6;9.0) |
| > 64 | 0.0 | 3.6 | 30.1 (1.0;502.4) | 0.0 | 22.3 | 5.0 (0.3;79.2) |

Prevalence ratio–indigenous/non-indigenous women.

Legend—< 25: women under 25; > 64: women over 64; LSIL: Low grande intra-epithelial lesion; HSIL+: High-grade intra-epithelial lesion or more severe; Indig: Indigenous women; Non-indig: Non-indigenous women; P–Prevalence; PR: Prevalence ratio; CI–Confidence Interval).

towards increasing the prevalence, although significantly only in non-indigenous women. It is possible to observe in the indigenous HSIL+ curve that there is a rapid increase in HSIL+ prevalence, reaching peaks in women from 25 to 34 years old, then a slight decrease, and a plateau after that.

## Discussion

As far as we noticed, this is the more extensive study regarding cervical cancer screening in indigenous women from Latin America. The quality of data allowed for a detailed analysis aiming to contribute to understanding the diseases' natural history and the impact of screening on indigenous from the Brazilian Amazon region. A higher prevalence of cytological LSIL and HSIl+ was observed among the indigenous, and the results in the screened population point to an urgent need to improve the current screening practice.

The evaluation of the results of the woman's first screening test (1st test) is related to the natural history of cancer. These women were not submitted to previous procedures in the cervix, so considering the high specificity of cytological tests [20, 23], the age-prevalence of results may reflect the course of the prevalent lesion.

We found LSIL results twice as prevalent in indigenous than in non-indigenous younger than 25 years. In the age group 40–64, it was 9.6 times more prevalent. Inferring that LSIL represents transient infections, we considered two hypotheses: indigenous women have more exposition to the HPV virus until older ages or the ability to eliminate the virus by indigenous

**Table 4. Prevalence ratio of cytological results in women attending screening or performing their first screening test, in indigenous and non-indigenous women.**

| | Indigenous | | Non-indigenous | |
|---|---|---|---|---|
| | LSIL | HSIL+ | LSIL | HSIL+ |
| | PR (CI 95%) | PR (CI 95%) | PR (CI 95%) | PR (CI 95%) |
| < 25 | 0.51 (0.16;1.62) | 0.49 (0.02;12.02) | 0.66 (0.59;0.73) | 1.65 (1.28;2.14) |
| 25 to 39 | 0.36 (0.11;1.17) | 0.77 (0.31;1.96) | 0.57 (0.44;0.74) | 0.59 (0.43;0.79) |
| 40 to 64 | 0.05 (0.00;0.81) | 0.84 (0.18;3.84) | 0.91 (0.39;2.13) | 0.19 (0.13;0.27) |
| > 64 | - | 0.56 (0.02;13.44) | 0.85 (0.05;14.43) | 0.10 (0.06;0.15) |

Prevalence ratio–screening test/first test.

Legend—< 25: women under 25; > 64: women over 64; LSIL: Low grande intra-epithelial lesion; HSIL+: High-grade intra-epithelial lesion or more severe; PR: Prevalence ratio; CI–Confidence Interval).

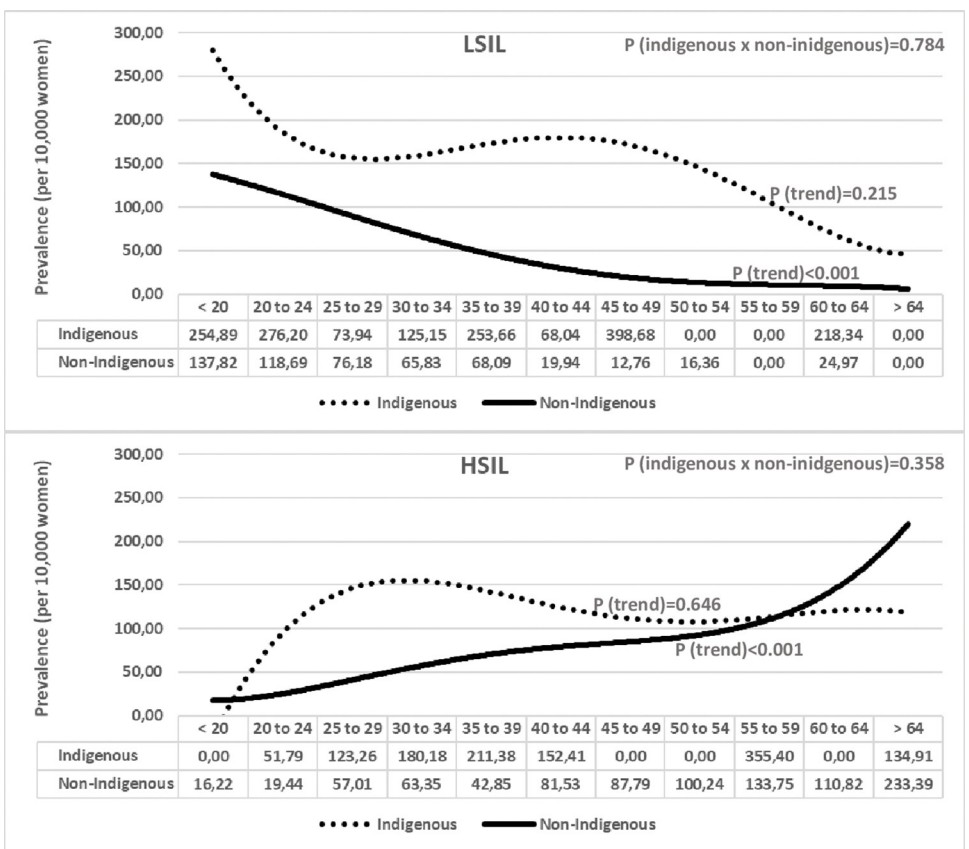

**Fig 1. Trends of prevalence of LSIL and HSIL+ cytological results in indigenous versus non-indigenous women, according to age groups.** P = P-value, by linear regression.

is compromised. Many studies report a higher prevalence of HPV in indigenous women [2, 8–10, 13, 15, 24]. In the Northern Brazilian Amazon, the HPV prevalence was reported at 39.7%, and it was more significant when more isolated the indigenous population [2]. The same authors found a high prevalence of high-risk HPV in indigenous women over 55, data corroborated by other studies [2, 10, 15, 24].

In high-income regions, the age-specific HPV prevalence curve points to a peak in women some years after sexual debut—around 20 years old, and a steady decline with aging related to the immunologic ability to eliminate the infection [25, 26]. A U-shaped prevalence curve is described in Latin American countries and Asia's lowest-income regions. In sub-Saharian African regions is observed a persistently high HPV prevalence with aging [26]. Those curves might be explained by persistent risk cofactors such as smoking, HIV, multiparity, etc. In our study, the indigenous results curves were more consistent with those HPV curves related to low-income regions.

The HPV prevalence genotype may also interfere. It is well-known that the time to HPV elimination varies according to the genotype [25]. Previous studies show a wide range of HPV subtypes in indigenous, the most prevalent types 6, 11, 16, 18, 31, 33, 45, 53, and 58 [2, 7, 11, 15, 24, 27–30]. The national vaccination program in Brazil introduced prophylaxis to HPV in 2014 and currently vaccinates girls and boys from 9 to 14 years old. The estimated coverage of indigenous in this target age is over 80% [31]. It is improbable women in this study may have been vaccinated. Considering the high LSIL prevalence in older indigenous women observed

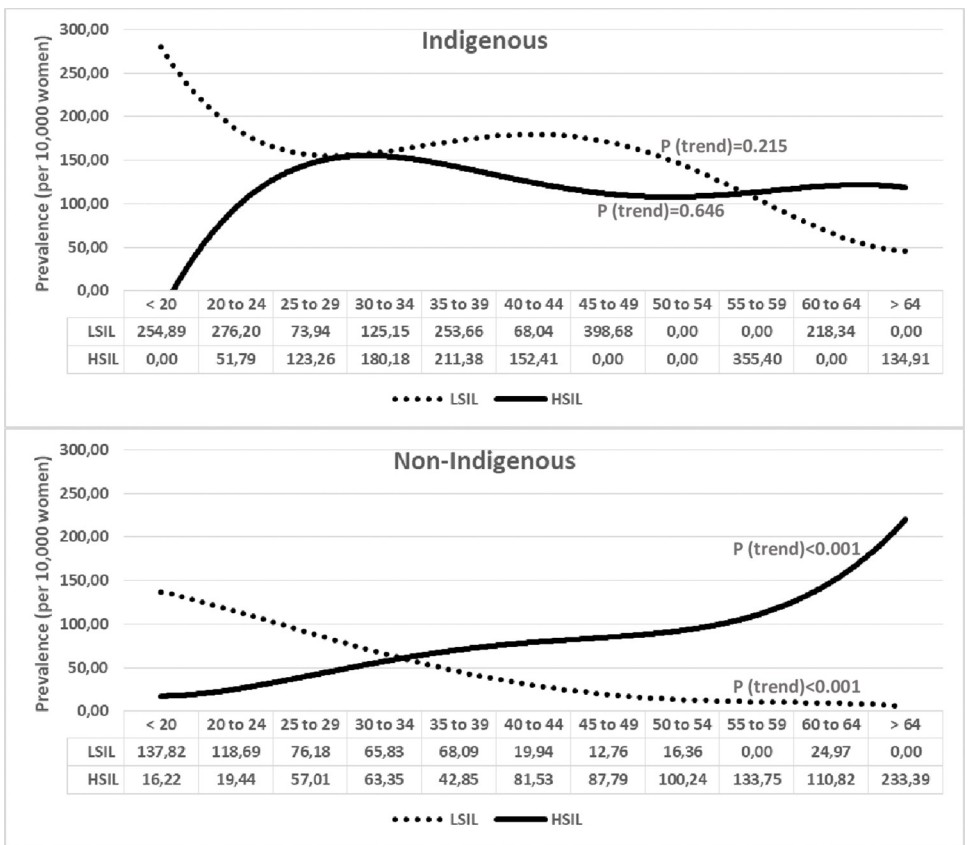

**Fig 2. Trends of prevalence of LSIL versus HSIL+ cytological results in indigenous and non-indigenous women, according to age groups.** P = P-value, by linear regression.

in this study, the known expected impact of the HPV vaccine, and the ability to reach high coverage vaccine rates, we believe our results should raise the debate to extend the HPV vaccine to older indigenous groups to avoid new infections and boost immunity.

We have found the prevalence of HSIL+ low in women younger than 25 in both groups (indigenous and non-indigenous). It supports the recommendation of the Brazilian Ministry of Health not to screen women before 25 years old. Speck *et al.* in 2015 claimed that indigenous women should start screening earlier, based on the finding of 0.5% of HSIL cytology result in indigenous women from Xingu, Brazilian Western-Center region, in women 12–24 years old [14]. We found the HSIL+ prevalence 24/10.000 tests in women under 25, which is lower than in the previous study. We must consider that our sample contemplates several ethnic indigenous groups from many regions of the Amazon rainforest, more representative of the indigenous people than the previous study. Nevertheless, we believe that even 0.5% is a shallow rate and does not justify anticipating screening, considering the high rates of HSIL regression in women under 30, more significant in women under 25 [32, 33]. The small benefit of screening at very younger ages should be counterbalanced by the preterm risks associated with identifying precursor lesions in this group [34].

HSIL+ was at least three times more frequent in indigenous women compared to non-indigenous in both 1st test (PR 3.0, 1.3;6.8 in women 25–39) and screening test (PR 4.0, 2.3;6.8 in women 25–39; PR 3.8, 1.6;9.0 in women 40–64). Although no study in Brazil compares indigenous versus non-indigenous women, some studies confirm the high prevalence of

precursor lesions in indigenous [2, 5, 7, 14]. In Northern Amazon, one study found an HSIL prevalence of 10.9% in Yanomami women and 2.5% in Macuxi and Wapishana women [2]. Interesting to note that Yanomami women is a group of recent contact with the non-indigenous population. The high LSIL prevalence also reinforces the previous arguments of immunologic decreased ability to eliminate the virus. It indicates that indigenous women are more vulnerable to developing precursor lesions. It highlights the need for specific recommendations on screening.

The screening effect was evaluated by comparing the results of screening tests versus 1[st] tests. In indigenous women, no effect of screening was observed in HSIL+ results. However, in non-indigenous women, as expected, a protective screening effect in preventing HSIL+ was observed in women over 25, more pronounced over 40: 80% protection in women aged 40 to 64 years and 90% in women over 64 years old.

One explanation for not observing the screening effect for HSIL+ in indigenous women is the limited sample. However, several factors contribute to the hypothesis that screening in the last decades in the indigenous population was of low efficiency. The first argument is the high prevalence of cervical cancer observed in indigenous women [3, 12, 35]. Renna-Junior *et al.* showed that indigenous women have two times more chance of being diagnosed in advanced stages when compared to the usual risk population [36]. Second, the evidence regarding screening coverage in indigenous women points to wide rate ranges [2, 7, 37–39], indicating irregular screening and non-standardized actions. Another attractive argument is indigenous behavior. Some authors point to sexual behavior in indigenous favoring cervical cancer risk factors [6, 10, 11, 27, 28, 36, 39–41]. Indeed, there are anthropologic barriers that might hamper willingness to screen. A study with indigenous oncologic patients found that 15% of cancer women refuse treatment [1].

We hypothesize that access to screening practices and treatment of precursor lesions is the main challenge to overcoming screening in indigenous women. In a simplistic model, currently, there are two demographic branches of the indigenous people: one living in indigenous territories in rainforest communities and another living in poor areas of urbanized regions. In the first scenario, indigenous health care is accessed through actions managed by governmental programs or by non-profit organizations. In these settings, the barriers are the region's geography, limiting health care, and non-sustainable activities.

Regarding the indigenous living in poor urban areas, those women accumulate the vulnerability related to social determinants, ethnicity, and the fragility of health systems observed in low-income regions—intersectionality. Both scenarios justify the high relevance of cervical cancer as a public health problem in indigenous women's health. The demographic register of the indigenous health system is a powerful tool to implement and deliver sustainable strategies when planning a specific cervical cancer screening program for this vulnerable population.

Our study has a great diversity of ethnicities of indigenous women from the Brazilian Amazon Rainforest. So, it may be representative of a great variety of indigenous populations. As far as we know, it is the larger sample of cytological results in the context of cervical cancer screening, and the information available makes it possible to trace the detailed profile of the screening characteristics of the population. Aggregated data should not be interpreted as an attempt to reduce the relevance of the ethnicity diversion but rather to support common strategies to guide public policies. Another strength is that the same high-quality laboratory has read all tests. The main limitation is the absence of follow-up in the results since a histological diagnosis would result in a more accurate figure of the natural history inferred. However, considering the paucity of studies on this population, our results should encourage and support control strategies to overcome the challenges.

## Conclusions

A higher prevalence of cytological LSIL and HSIl+ was observed among indigenous women. The protective screening effect in reducing HSIL+ prevalence was not observed in indigenous. The results suggest evaluating the vaccine age group and an urgent need to improve access to screening practices in Brazilian indigenous women.

## Supporting information

**S1 Table. Description of health expeditions with screening activities in indigenous women from the Brazilian Amazon from 2007 to 2019.**
(DOCX)

## Author Contributions

**Conceptualization:** Iria Ribeiro Novais, Luiz Carlos Zeferino, Diama Bhadra Vale.

**Data curation:** Iria Ribeiro Novais, Camila Olegario Coelho, Helymar Costa Machado, Diama Bhadra Vale.

**Formal analysis:** Iria Ribeiro Novais, Helymar Costa Machado, Diama Bhadra Vale.

**Funding acquisition:** Diama Bhadra Vale.

**Investigation:** Iria Ribeiro Novais, Diama Bhadra Vale.

**Methodology:** Iria Ribeiro Novais, Helymar Costa Machado, Diama Bhadra Vale.

**Project administration:** Diama Bhadra Vale.

**Resources:** Diama Bhadra Vale.

**Supervision:** Diama Bhadra Vale.

**Validation:** Fernanda Surita, Diama Bhadra Vale.

**Visualization:** Diama Bhadra Vale.

**Writing – original draft:** Iria Ribeiro Novais, Camila Olegario Coelho, Diama Bhadra Vale.

**Writing – review & editing:** Iria Ribeiro Novais, Camila Olegario Coelho, Helymar Costa Machado, Fernanda Surita, Luiz Carlos Zeferino, Diama Bhadra Vale.

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
