## [Decision Letter · Decision Letter 0]

5 Jul 2023

PONE-D-23-11232Cervical Cancer Screening in Brazilian Amazon Indigenous Women: a wakeup call to expand vaccination and improve access to cervical cancer prevention practicesPLOS ONE

Dear Dr. Vale,

Thank you for submitting your manuscript to PLOS ONE. After careful consideration, we feel that it has merit but does not fully meet PLOS ONE’s publication criteria as it currently stands. Therefore, we invite you to submit a revised version of the manuscript that addresses the points raised during the review process.

We look forward to receiving your revised manuscript.

Kind regards,

Antonio Carlos Rosario Vallinoto, Ph.D.

Academic Editor

PLOS ONE

“IRN was funded by a grant from the "Coordenação de Aperfeiçoamento de Pessoal de Nível Superior" (CAPES) for her Master's degree.”

3. We noted in your submission details that a portion of your manuscript may have been presented or published elsewhere. [The manuscript is included in a Master Thesis to be defended and published on the University of Campinas website. Results were accepted at the International Cancer Screening Network (ICSN) 2023, to be helded in June 2023.] Please clarify whether this publication was peer-reviewed and formally published. If this work was previously peer-reviewed and published, in the cover letter please provide the reason that this work does not constitute dual publication and should be included in the current manuscript.

Reviewers' comments:

Reviewer's Responses to Questions

**Comments to the Author**

1. Is the manuscript technically sound, and do the data support the conclusions?

Reviewer #1: Partly

Reviewer #2: Yes

Reviewer #3: Partly

2. Has the statistical analysis been performed appropriately and rigorously? 

Reviewer #1: Yes

Reviewer #2: Yes

Reviewer #3: Yes

3. Have the authors made all data underlying the findings in their manuscript fully available?

Reviewer #1: Yes

Reviewer #2: No

Reviewer #3: No

4. Is the manuscript presented in an intelligible fashion and written in standard English?

Reviewer #1: Yes

Reviewer #2: Yes

Reviewer #3: No

5. Review Comments to the Author

Reviewer #1: The manuscript presents unpublished data regarding screening tests for cervical cancer in included indigenous and non-indigenous women, with a considerable sample number. It has scientific and sacred merit in public health. However, it is important to review some points to know:

Title – The data presented do not support the inference of expansion of vaccination. It would be interesting to adjust and place: an alteration for the intensification of public policies for the prevention of CC.

The study does not present data related to the occurrence of CC in the investigated populations.

Methodology:

1) What is the Nomenclature system for cytopathological reports adopted in the Program? Bethesda or Brazilian Nomenclature for Cytopathological Reports? It is necessary to insert this information in the methodology.

2) The following outcome variables were used: LSIL, HSIL, squamous cell carcinoma, atypical glandular cells, adenocarcinoma in situ and invasive adenocarcinoma and smears without cytological alterations or normal. However, the frequency of these events was not described in the results or in the tables.

Results

1) What is the minimum and maximum age of the participants involved in the study? It is necessary to insert this information in the text.

2) Were there cases of neoplasia in the investigated populations?

3) Table 01 presents only the cytological results (low and high grade lesion) according to age group. It does not present the general characteristics of the examined population. Likewise, it does not inform all the variables considered as an outcome, which were described in the methodology (LSIL, HSIL, Squamous cell carcinoma, atypical glandular cells, adenocarcinoma in situ and invasive adenocarcinoma and smears without cytological changes or normal). These data should all be considered in epidemiology studies and in cervical cancer screening.

4) What are the prevalence rates of cytological changes in the group of women who were screened for cervical cancer? Table 2 does not describe raw frequency or prevalence data for these outcomes.

5) In the Tables, it is necessary to insert the raw prevalence data in the investigated population, the prevalence rate and also the “p” value obtained with the statistical analysis.

Discussion

1) 1st Paragraph - The data collected in the present study are not enough to understand the natural history of the disease, as they do not provide information about the follow-up of the participants, evolution or regression of the lesions, therapeutic interventions, nor do they consider the intrinsic factors of the host . It is necessary to adjust/remove this statement in the discussion.

2) 3rd Paragraph - The prevalence rate is not enough to assess the magnitude of the risk, which is achieved through odds ratio. Thus, it is not possible to state that the prevalence is twice as high, but rather that there is a greater probability of occurrence of the event (LSIL) in indigenous women, as PR values above 1 demonstrate a positive association between the events. Therefore, I reinforce the idea that it is necessary to adjust tables 02 and 03, with the insertion of data on prevalence, prevalence rates and p value. As well as the statistical method employed.

3) 4th Paragraph - The study did not aim to evaluate the prevalence of HPV in the examined groups and its correlation with the development of intraepithelial lesions. Thus, it is important to consider in the discussion aspects related to prevalence rates of cytological results (LSIL and HSIl) in the general population and in indigenous populations obtained in other studies conducted in Brazil and in other countries.

4) 5th Paragraph – It is not possible to infer the protective effect of carrying out the screening test for cervical cancer screening in the present study, as there is no description of the cases of cancer in the examined group, there was follow-up of patients to assess the progression of high-grade lesions to cancer.

Reviewer #2: The manuscript “Cervical Cancer Screening in Brazilian Amazon Indigenous Women: a wakeup call to expand vaccination and improve access to cervical cancer prevention practices” by Novais and collaborators, aims to evaluate the prevalence profile of cytological results of cervical cancer screening in Brazilian indigenous women by age and frequency of examinations compared to non-indigenous women. For this purpose, tests performed on indigenous women assisted in the Brazilian Amazon from 2007 to 2019 (3,231 tests) were used, which were compared with the results observed in non-indigenous women (698,415 tests) in the same period.

It is an important study from an epidemiological and public health point of view, whose results will serve to reinforce and support the adoption of more appropriate public health policies aimed at health care for indigenous peoples in Brazil. It therefore deserves to be published.

The study addresses indigenous peoples who exhibit remarkable biological, linguistic and cultural diversity, directly related to the origins and evolutionary events they experienced during their migration in the American continent, and who currently have different socioeconomic realities resulting from contact with non-indigenous people. In this context, analyzing the results globally is obviously important as a parameter or indicator of the health status of indigenous women, but in my view it fails to address the impact of local (regional), ethnic (biological) and sociocultural realities on the prevalence of changes cytology in indigenous people. In the Amazon there are at least 170 indigenous peoples or ethnic groups, the majority living in villages, but with different socioeconomic realities resulting from contact with non-indigenous people, as mentioned above, and a fraction living in urban contexts, generally subjected to macro-social changes imposed by life in cities, with changes in the traditional indigenous lifestyle, which obviously has an impact on their patterns of illness.

Therefore, I consider it essential to analyze the data separately by ethnicity or indigenous people, as I imagine that this information is available in the project's database. This is very important even to assess the efficiency of health care in the different Special Indigenous Health Districts (DSEI), which are the units responsible for health care for indigenous peoples, regionally. In the same way, I consider it important to analyze the data separating indigenous villagers from urban indigenous people, if applicable.

Another question refers to the “wakeup call to expand vaccination”, because, unless I'm mistaken, no molecular test was carried out to verify the circulating HPV serotypes in the indigenous people, and whether these types are being covered by the vaccine that is offered to the indigenous people. There is also no data on the vaccination coverage of the indigenous people described in the work, if they were vaccinated, how many doses they received, and the age groups contemplated with the vaccination. These are data that I consider important to help in the evaluation of vaccination against HPV and its expansion, if applicable, in indigenous women.

I also suggest checking the nomenclature of the cytological results according to what is recommended by the Brazilian nomenclature for cervical cytopathological results in relation to the description of HSIL+. Still on this, I consider it inappropriate to use abbreviations in the abstract of the work, as it makes it difficult for the reader who is not familiar with the subject to understand.

Another minor and punctual question refers to considering that “Indigenous women are vulnerable to cervical cancer”. It seems to me that it is a question of assessing whether they are more or less vulnerable than non-indigenous women, since they are also vulnerable, as far as I know, and under what conditions does indigenous women become more or less vulnerable, which involves aspects biological and sociocultural and environmental.

In principle, these are my comments and contributions on the manuscript.

Reviewer #3: The article seeks an important approach for proposing and monitoring strategies to expand vaccination and improve access to cervical cancer prevention practices. It presents a large number of indigenous and non-indigenous women enrolled and compares the prevalence of initial and more advanced lesions in the progression to cervical cancer, also stratifying by age group.

Some adjustments are necessary to give greater coherence to the objectives and conclusions of the article. The most important of these is the lack of information about vaccination of women enrolled as participants. It is not possible to accurately discuss prevention and expansion of vaccination when specific information about vaccination on the study population is not known.

As the authors group more advanced lesions (HSIL+), the terms “cervical cancer” and “cervical intraepithelial neoplasia” lose relevance and should not be included as keywords.

It is not possible to define the rigor in the classification of “indigenous”, since the authors do not define which areas and the criteria that define them as “remote areas of the Amazon rainforest”. In the discussion, we found the information that “there are two demographic branches of the indigenous people: one living in indigenous territories in rainforest communities and another living in poor areas of urbanized regions“. It would be interesting to carry out the analysis by subcategorizing these two demographic branches. The addition of a map identifying the studied populations is strongly suggested.

It is recommended to quantitatively list the number of women excluded from each group by each exclusion criterion.

Data analyzed without vaccination information do not support the conclusion “The results suggest expanding the vaccine age group and an urgent need to improve access to screening practices in Brazilian indigenous women.”

In “Availability of data and materials”, replace “availabel” with “available”.

In the legend of tables 1 and 2, replace “Non-indigenous women” with “Non-indig: Non-indigenous women”.

It is necessary to standardize the terms “1st test” and “screening test”, in order to avoid confusion. Sometimes we find “non-first test”, “test for the first time”, “first screening test”, “first cervical cancer screening test”, “cancer screening”.

In the legend of Table 4, it is suggested to remove “Indig: Indigenous women; Non-indigenous women; P – Prevalence;”.

6. PLOS authors have the option to publish the peer review history of their article (what does this mean?). If published, this will include your full peer review and any attached files.

Reviewer #1: No

Reviewer #2: No

Reviewer #3: **Yes: **Igor Brasil-Costa

---

## [Author Response · Author response to Decision Letter 0]

2 Oct 2023

Response to reviewers

PONE-D-23-11232

 Dear Editor and Reviewers. 

 Thanks for the opportunity to review the manuscript. Your valuable comments and corrections improved its quality.

 The answers to the queries are in the following text.

 Best regards,

 The corresponding author.

1) Please ensure that your manuscript meets PLOS ONE's style requirements, including those for file naming. The PLOS ONE style templates can be found at https://journals.plos.org/plosone/s/file?id=wjVg/PLOSOne_formatting_sample_main_body.pdf and https://journals.plos.org/plosone/s/file?id=ba62/PLOSOne_formatting_sample_title_authors_affiliations.pdf

R: Adjusted.

2) Thank you for stating the following in the Acknowledgments Section of your manuscript: "IRN was funded by a grant from the "Coordenação de Aperfeiçoamento de Pessoal de Nível Superior" (CAPES) for her Master's degree." We note that you have provided funding information that is not currently declared in your Funding Statement. However, funding information should not appear in the Acknowledgments section or other areas of your manuscript. We will only publish funding information present in the Funding Statement section of the online submission form. Please remove any funding-related text from the manuscript and let us know how you would like to update your Funding Statement. Currently, your Funding Statement reads as follows: "The author(s) received no specific funding for this work." Please include your amended statements within your cover letter; we will change the online submission form on your behalf.

R: Dear editor. The support by CAPES was made as a scholarship to the first author during her Master's. It is true that "The author(s) received no specific funding for this work." There is no direct funding for the study. The national agency for post-graduation support (CAPES) requires the following statement in the Acknowledgments section on every paper published by a post-graduation student: "This study was financed in part by the Coordenação de Aperfeiçoamento de Pessoal de Nível Superior - Brasil (CAPES) - Finance Code 001". We will change this information for the Funding Statement section. 

3) We noted in your submission details that a portion of your manuscript may have been presented or published elsewhere. [The manuscript is included in a Master Thesis to be defended and published on the University of Campinas website. Results were accepted at the International Cancer Screening Network (ICSN) 2023, to be helded in June 2023.] Please clarify whether this publication was peer-reviewed and formally published. If this work was previously peer-reviewed and published, in the cover letter please provide the reason that this work does not constitute dual publication and should be included in the current manuscript. 

R: Some of the results of this study were presented as a poster at the ICSN Conference. The Conference did not publish the Anais. The manuscript in its original form was in the results section of the Master Thesis of IRN, as one of the products of her Master's. In our post-graduation program, a submitted paper is a condition for the Master's defense. The Master's was defended in May, and the University stores all the thesis copies in an online repository after an embargo of one year. It was not peer-reviewed but will be formally published on the website as the entire thesis AFTER MAY 2024. As usual, if the paper is accepted, we will submit a permission statement before it. We added that information to the cover letter.

4) We note that you have indicated that data from this study are available upon request. PLOS only allows data to be available upon request if there are legal or ethical restrictions on sharing data publicly. For more information on unacceptable data access restrictions, please see http://journals.plos.org/plosone/s/data-availability#loc-unacceptable-data-access-restrictions. In your revised cover letter, please address the following prompts: a) If there are ethical or legal restrictions on sharing a de-identified data set, please explain them in detail (e.g., data contain potentially sensitive information, data are owned by a third-party organization, etc.) and who has imposed them (e.g., an ethics committee). Please also provide contact information for a data access committee, ethics committee, or other institutional body to which data requests may be sent. b) If there are no restrictions, please upload the minimal anonymized data set necessary to replicate your study findings as either Supporting Information files or to a stable, public repository and provide us with the relevant URLs, DOIs, or accession numbers. For a list of acceptable repositories, please see http://journals.plos.org/plosone/s/data-availability#loc-recommended-repositories. We will update your Data Availability statement on your behalf to reflect the information you provide.

R: We deposited metadata at https://doi.org/10.25824/redu/9BLFCK, and added this information on the manuscript.

5) Your ethics statement should only appear in the Methods section of your manuscript. If your ethics statement is written in any section besides the Methods, please delete it from any other section.

R: Corrected.

Reviewer #1

1) Title – The data presented do not support the inference of expansion of vaccination. It would be interesting to adjust and place: an alteration for the intensification of public policies for the prevention of CC. The study does not present data related to the occurrence of CC in the investigated populations.

R: Thanks for the suggestion. We replaced by "towards the intensification of public policies for the prevention."

2) Methodology: 1) What is the Nomenclature system for cytopathological reports adopted in the Program? Bethesda or Brazilian Nomenclature for Cytopathological Reports? It is necessary to insert this information in the methodology.

R: We clarified this issue in methoods section – "The outcome variables were the cytological results. Cytopathological reports used in the laboratory follows the Brazilian Nomenclature for Cytopathological Reports, based on the Bethesda System (ref). In this study we categoryzed results as "Low-grade Squamous Intraepithelial Lesion (LSIL)…"

3) Methodology: 2) The following outcome variables were used: LSIL, HSIL, squamous cell carcinoma, atypical glandular cells, adenocarcinoma in situ and invasive adenocarcinoma and smears without cytological alterations or normal. However, the frequency of these events was not described in the results or in the tables.

R: We only use the three categories – LSIL, HSIL+, and total as variables. We agree that this information was not clear throughout the methods and results. We adjusted the text on methods: "In this study, we categorized results as "Low-grade Squamous Intraepithelial Lesion (LSIL)" or "High-grade Squamous Intraepithelial Lesion or more severe" (HSIL+). HSIL+ corresponded to Squamous Cell Carcinoma, Atypical Glandular Cells, Adenocarcinoma in situ, and Invasive Adenocarcinoma. Because of their low frequency, we grouped those results into one single category."; and in the results, we changed the introduction of Table 1: "Table 1 shows the proportions of the aggregated results (LSIL, HSIL+, and totals)."

4) Results - 1) What is the minimum and maximum age of the participants involved in the study? It is necessary to insert this information in the text.

R: According to the Ethics Committee, the minimum age of indigenous women screened is sensitive data. We displayed only aggregated data. Screening in Brazil is opportunistic, so, despite the recommendation of screening women over 25, no control prevents a healthcare worker from screening younger women. In fact, as we stated in the discussion, previous studies recommended anticipating screening in indigenous women.

5) Results - 2) Were there cases of neoplasia in the investigated populations?

R: We completed the information on methods: "Tests results were returned to PHC, and abnormal results followed the regular multilevel pathway throughout referral regional facilities to colposcopy and treatment, including results suggestive of invasive lesions. In indigenous women, only one test suggested invasive lesion."

6) Results - 3) Table 01 presents only the cytological results (low and high grade lesion) according to age group. It does not present the general characteristics of the examined population. Likewise, it does not inform all the variables considered as an outcome, which were described in the methodology (LSIL, HSIL, Squamous cell carcinoma, atypical glandular cells, adenocarcinoma in situ and invasive adenocarcinoma and smears without cytological changes or normal). These data should all be considered in epidemiology studies and in cervical cancer screening.

R: We answered these issues previously with some adjustments to the text. We only use the three categories – LSIL, HSIL+, and total as variables. We agree that this information was not clear throughout the methods and results. We adjusted the text on methods.

7) Results - 4) What are the prevalence rates of cytological changes in the group of women who were screened for cervical cancer? Table 2 does not describe raw frequency or prevalence data for these outcomes.

R: In tables 2 ad 3, prevalences are presented in the columns "P" by 10,000 tests. We choose to present data as prevalence instead of frequencies because we believe prevalence is a stronger indicator and demonstrates the load of risk to the data.

8) Results - 5) In the Tables, it is necessary to insert the raw prevalence data in the investigated population, the prevalence rate and also the "p" value obtained with the statistical analysis.

R: We stated on methods that we choose to present data as prevalence rate ("P" in the tables). We did not display the raw figures of the test because we think it is not as informative as prevalence data, which is much more sophisticated as a risk indicator. The p-value was calculated in the regression analysis presented in the figures. To compare prevalence rates, we choose the prevalence ratio with confidence interval as a measure of risk (PR (95% CI).

9) Discussion - 1) 1st Paragraph - The data collected in the present study are not enough to understand the natural history of the disease, as they do not provide information about the follow-up of the participants, evolution or regression of the lesions, therapeutic interventions, nor do they consider the intrinsic factors of the host . It is necessary to adjust/remove this statement in the discussion.

R: Thanks for helping us to clarify this issue. It was not our intention to claim our data as the corn data regarding the natural history of the disease, but only to contribute and support the knowledge regarding this issue. But we strongly believe in the importance of this data in face of the scarcity of the literature in this setting. We agree the writing may have given a wrong impression. We made adjustments throughout the text. Regarding the 1st paragrapf of the discussion, we wrote: "The quality of data allowed for a detailed analysis AIMING TO CONTRIBUTE TO THE UNDERSTANDING of the diseases' natural history and the impact of screening on indigenous from the Brazilian Amazon region."

10) Discussion - 2) 3rd Paragraph - The prevalence rate is not enough to assess the magnitude of the risk, which is achieved through odds ratio. Thus, it is not possible to state that the prevalence is twice as high, but rather that there is a greater probability of occurrence of the event (LSIL) in indigenous women, as PR values above 1 demonstrate a positive association between the events. Therefore, I reinforce the idea that it is necessary to adjust tables 02 and 03, with the insertion of data on prevalence, prevalence rates and p value. As well as the statistical method employed. 

R: We did not use the prevalence rate to assess the magnitude of risk when comparing the prevalence rates. We used PREVALENCE RATIO with confidence intervals – the statistical method employed, a well-known measure of risk in clinical epidemiology. In methods, we described the statistics used in the study. 

11) Discussion - 3) 4th Paragraph - The study did not aim to evaluate the prevalence of HPV in the examined groups and its correlation with the development of intraepithelial lesions. Thus, it is important to consider in the discussion aspects related to prevalence rates of cytological results (LSIL and HSIl) in the general population and in indigenous populations obtained in other studies conducted in Brazil and in other countries.

R: We conducted a systematic review regarding indigenous women in Latin America and found no manuscript comparing non-indigenous and indigenous prevalence. We included in the discussion: "Although no study in Brazil compares indigenous versus non-indigenous women, some studies confirm the high prevalence of precursor lesions in indigenous [2,5,7,14]. In Northern Amazon, one study found an HSIL prevalence of 10.9% in Yanomami women and 2.5% in Macuxi and Wapishana women (Fonseca). Interesting to note that Yanomami women are a group of recent contact with the non-indigenous population. The high LSIL prevalence also reinforces the previous arguments of immunologic decreased ability to eliminate the virus."

12) Discussion - 4) 5th Paragraph – It is not possible to infer the protective effect of carrying out the screening test for cervical cancer screening in the present study, as there is no description of the cases of cancer in the examined group, there was follow-up of patients to assess the progression of high-grade lesions to cancer.

R: Long-term outcomes are very difficult to achieve and is possible to use intermediary end-points as surrogates. Our aim was to evaluate the screening actions in the context of giving support to implementing public policies. In the literature there are several examples of audits and recommendations using surrogates to evaluate screening programs. We modified on the citied paragraph: "However, in non-indigenous women, as expected, a protective screening effect in preventing HSIL+ was observed in women over 25, more pronounced over 40… "

Reviewer #2

1) It is an important study from an epidemiological and public health point of view, whose results will serve to reinforce and support the adoption of more appropriate public health policies aimed at health care for indigenous peoples in Brazil. 

R: Thanks for the important comment. It is in line with our thoughts.

2) The study addresses indigenous peoples who exhibit remarkable biological, linguistic and cultural diversity, directly related to the origins and evolutionary events they experienced during their migration in the American continent, and who currently have different socioeconomic realities resulting from contact with non-indigenous people. In this context, analyzing the results globally is obviously important as a parameter or indicator of the health status of indigenous women, but in my view it fails to address the impact of local (regional), ethnic (biological) and sociocultural realities on the prevalence of changes cytology in indigenous people. In the Amazon there are at least 170 indigenous peoples or ethnic groups, the majority living in villages, but with different socioeconomic realities resulting from contact with non-indigenous people, as mentioned above, and a fraction living in urban contexts, generally subjected to macro-social changes imposed by life in cities, with changes in the traditional indigenous lifestyle, which obviously has an impact on their patterns of illness. Therefore, I consider it essential to analyze the data separately by ethnicity or indigenous people, as I imagine that this information is available in the project's database. This is very important even to assess the efficiency of health care in the different Special Indigenous Health Districts (DSEI), which are the units responsible for health care for indigenous peoples, regionally. In the same way, I consider it important to analyze the data separating indigenous villagers from urban indigenous people, if applicable.

R: We totally agree with your comments and it was one of our objectives while writing the study protocol. However, when analysing it we found out we would fail in qualifying the analysis when limiting the samples. In fact, in a systematic review we are conducting about the topic, we observe that the majority of the papers present data by ethnicity, and those results are fragile to support hypothesis and public policies (small sample). We realize that grouping results would give more important information and the robusteness of the analysis would make the recommendations stronger. We added on the limitation: "Aggregated data should not be interpreted as an attempt to reduce the relevance of the ethnicity diversion, but rather to support common strategies to guide public policies."

3) Another question refers to the "wakeup call to expand vaccination", because, unless I'm mistaken, no molecular test was carried out to verify the circulating HPV serotypes in the indigenous people, and whether these types are being covered by the vaccine that is offered to the indigenous people. There is also no data on the vaccination coverage of the indigenous people described in the work, if they were vaccinated, how many doses they received, and the age groups contemplated with the vaccination. These are data that I consider important to help in the evaluation of vaccination against HPV and its expansion, if applicable, in indigenous women.

R: Thanks for your comment and perspective. We added in the discussion: "The national vaccination program in Brazil introduced prophylaxis to HPV in 2014 and currently vaccinates girls and boys from 9 to 14 years old. The estimated coverage of indigenous in this target age is over 80% [31]. It is improbable women in this study may have been vaccinated. Considering the high LSIL prevalence in older indigenous women observed in this study, the known expected impact of the HPV vaccine, and the infrastructure and ability to reach high coverage vaccine rates, we believe our results should raise the debate to extend HPV vaccine to older indigenous groups to avoid new infections and boost immunity."

4) I also suggest checking the nomenclature of the cytological results according to what is recommended by the Brazilian nomenclature for cervical cytopathological results in relation to the description of HSIL+. 

R: We agree it was not clear in the methods. As also commented by Reviewer 1, we adjusted the text on methods: "In this study, we categorized results as "Low-grade Squamous Intraepithelial Lesion (LSIL)" or "High-grade Squamous Intraepithelial Lesion or more severe" (HSIL+). HSIL+ corresponded to Squamous Cell Carcinoma, Atypical Glandular Cells, Adenocarcinoma in situ, and Invasive Adenocarcinoma. Because of their low frequency, we grouped those results into one single category."

5) Still on this, I consider it inappropriate to use abbreviations in the abstract of the work, as it makes it difficult for the reader who is not familiar with the subject to understand.

R: Adjusted.

6) Another minor and punctual question refers to considering that "Indigenous women are vulnerable to cervical cancer". It seems to me that it is a question of assessing whether they are more or less vulnerable than non-indigenous women, since they are also vulnerable, as far as I know, and under what conditions does indigenous women become more or less vulnerable, which involves aspects biological and sociocultural and environmental.

R: We agree and adjusted the text (introduction): "Indigenous women in Latin America are more vulnerable to cervical cancer than non-indigenous women."

Reviewer #3

1) Some adjustments are necessary to give greater coherence to the objectives and conclusions of the article. The most important of these is the lack of information about vaccination of women enrolled as participants. It is not possible to accurately discuss prevention and expansion of vaccination when specific information about vaccination on the study population is not known.

R: Thanks for the valuable comment. We were more discreet regarding the vaccination aspects throughout the text and, as also suggested by reviewer 2, we added to the discussion: "The national vaccination program in Brazil introduced prophylaxis to HPV in 2014 and currently vaccinates girls and boys from 9 to 14 years old. The estimated coverage of indigenous in this target age is over 80% [31]. It is improbable women in this study may have been vaccinated. Considering the high LSIL prevalence in older indigenous women observed in this study, the known expected impact of the HPV vaccine, and the infrastructure and ability to reach high coverage vaccine rates, we believe our results should raise the debate to extend HPV vaccine to older indigenous groups to avoid new infections and boost immunity."

2) As the authors group more advanced lesions (HSIL+), the terms "cervical cancer" and "cervical intraepithelial neoplasia" lose relevance and should not be included as keywords.

R: We choose to keep those terms because the study refers to those general topics rather than being related to these specific outcomes.

3) It is not possible to define the rigor in the classification of "indigenous", since the authors do not define which areas and the criteria that define them as "remote areas of the Amazon rainforest". In the discussion, we found the information that "there are two demographic branches of the indigenous people: one living in indigenous territories in rainforest communities and another living in poor areas of urbanized regions ". It would be interesting to carry out the analysis by subcategorizing these two demographic branches. The addition of a map identifying the studied populations is strongly suggested.

R: Thanks for noting. We added a table describing the expeditions and ethnicities covered. On results: "The description of the expeditions and ethnicities are displayed on S1 table." 

4) It is recommended to quantitatively list the number of women excluded from each group by each exclusion criterion.

R: In the methods it was added: In the original collection form, "frequency" was recorded as "first test," "less than one year," "one year," "two years,"…, "five years," and "more than five years." After exclusion criteria were applied, in the category "frequency," those with unknown "frequency" were redistributed by age group and result, and then excluded those reported as "less than one year" because they were considered as "tests collected for purposes other than screening." And on the results it was added: From the total number of tests, those were excluded in indigenous and non-indigenous women, respectively: hysterectomy 19 and 21,231 tests; tests collected for purposes other than screening 222 and 60,780 tests; unsatisfactory 38 and 7,394 tests; and without information on age 7 and 418 tests; or result 38 and 7,878 tests. More than one exclusion criteria could be applied to each test.

5) Data analyzed without vaccination information do not support the conclusion "The results suggest expanding the vaccine age group and an urgent need to improve access to screening practices in Brazilian indigenous women."

R: We agree we should be more discreet in the conclusion. We added the information on vaccination to the discussion as previously commented and changed the sentence to: "the results suggest EVALUATING the vaccine age group…" 

6) In "Availability of data and materials", replace "availabel" with "available".

R: Thanks for noting. Corrected.

7) In the legend of tables 1 and 2, replace "Non-indigenous women" with "Non-indig: Non-indigenous women".

R: Corrected.

8) It is necessary to standardize the terms "1st test" and "screening test", in order to avoid confusion. Sometimes we find "non-first test", "test for the first time", "first screening test", "first cervical cancer screening test", "cancer screening".

R: Thanks for noting. We went through all the documents adjusting the terms.

9) In the legend of Table 4, it is suggested to remove "Indig: Indigenous women; Non-indigenous women; P – Prevalence;".

R: Corrected.

---

## [Decision Letter · Decision Letter 1]

13 Nov 2023

Cervical cancer screening in Brazilian Amazon Indigenous women: towards the intensification of public policies for prevention

PONE-D-23-11232R1

Dear Dr. Vale,

We’re pleased to inform you that your manuscript has been judged scientifically suitable for publication and will be formally accepted for publication once it meets all outstanding technical requirements.

Kind regards,

Antonio Carlos Rosario Vallinoto, Ph.D.

Academic Editor

PLOS ONE

Additional Editor Comments (optional):

Reviewers' comments:

Reviewer's Responses to Questions

**Comments to the Author**

1. If the authors have adequately addressed your comments raised in a previous round of review and you feel that this manuscript is now acceptable for publication, you may indicate that here to bypass the “Comments to the Author” section, enter your conflict of interest statement in the “Confidential to Editor” section, and submit your "Accept" recommendation.

Reviewer #1: All comments have been addressed

Reviewer #2: All comments have been addressed

Reviewer #3: (No Response)

2. Is the manuscript technically sound, and do the data support the conclusions?

Reviewer #1: Yes

Reviewer #2: Yes

Reviewer #3: (No Response)

3. Has the statistical analysis been performed appropriately and rigorously? 

Reviewer #1: Yes

Reviewer #2: Yes

Reviewer #3: (No Response)

4. Have the authors made all data underlying the findings in their manuscript fully available?

Reviewer #1: No

Reviewer #2: Yes

Reviewer #3: (No Response)

5. Is the manuscript presented in an intelligible fashion and written in standard English?

Reviewer #1: Yes

Reviewer #2: Yes

Reviewer #3: (No Response)

6. Review Comments to the Author

Reviewer #1: (No Response)

Reviewer #2: The authors adequately addressed my comments and suggestions made previously and I consider that the manuscript is acceptable for publication, providing important data on a prevalence profile of cytological exam results in cervical cancer screening in indigenous women, which will contribute important information for the adoption of more appropriate public health policy methods for health care for Brazilian indigenous women.

Reviewer #3: (No Response)

7. PLOS authors have the option to publish the peer review history of their article (what does this mean?). If published, this will include your full peer review and any attached files.

Reviewer #1: **Yes: **Jacqueline Cortinhas Monteiro

Reviewer #2: No

Reviewer #3: **Yes: **Igor Brasil-Costa

---

## [Editor Report · Acceptance letter]

1 Dec 2023

PONE-D-23-11232R1 

Cervical cancer screening in Brazilian Amazon Indigenous women: towards the intensification of public policies for prevention. 

Dear Dr. Vale:

I'm pleased to inform you that your manuscript has been deemed suitable for publication in PLOS ONE. Congratulations! Your manuscript is now with our production department. 

Kind regards, 

on behalf of

Dr. Antonio Carlos Rosario Vallinoto 

Academic Editor

PLOS ONE